# Exploiting the Symmetry of Integral Transforms for Featuring Anuran Calls

**Amalia Luque** [1,*] , **Jesús Gómez-Bellido** [1] , **Alejandro Carrasco** [2] **and Julio Barbancho** [3]

1  Ingeniería del Diseño, Escuela Politécnica Superior, Universidad de Sevilla, 41004 Sevilla, Spain; jesgombel@outlook.es
2  Tecnología Electrónica, Escuela Ingeniería Informática, Universidad de Sevilla, 41004 Sevilla, Spain; acarrasco@us.es
3  Tecnología Electrónica, Escuela Politécnica Superior, Universidad de Sevilla, 41004 Sevilla, Spain; jbarbancho@us.es
*  Correspondence: amalialuque@us.es; Tel.: +34-955-420-187

**Abstract:** The application of machine learning techniques to sound signals requires the previous characterization of said signals. In many cases, their description is made using cepstral coefficients that represent the sound spectra. In this paper, the performance in obtaining cepstral coefficients by two integral transforms, Discrete Fourier Transform (DFT) and Discrete Cosine Transform (DCT), are compared in the context of processing anuran calls. Due to the symmetry of sound spectra, it is shown that DCT clearly outperforms DFT, and decreases the error representing the spectrum by more than 30%. Additionally, it is demonstrated that DCT-based cepstral coefficients are less correlated than their DFT-based counterparts, which leads to a significant advantage for DCT-based cepstral coefficients if these features are later used in classification algorithms. Since the DCT superiority is based on the symmetry of sound spectra and not on any intrinsic advantage of the algorithm, the conclusions of this research can definitely be extrapolated to include any sound signal.

**Keywords:** spectrum symmetry; DCT; MFCC; audio features; anuran calls

## 1. Introduction

Automatic processing of sound signals is a very active topic in many fields of science and engineering which find applications in multiple areas, such as speech recognition [1], speaker identification [2,3], emotion recognition [4], music classification [5], outlier detection [6], classification of animal species [7–9], detection of biomedical disease [10], and design of medical devices [11]. Sound processing is also applied in urban and industrial contexts, such as environmental noise control [12], mining [13], and transportation [14,15].

These applications typically include, among their first steps, the characterization of the sound: a process which is commonly known as feature extraction [16]. A recent survey of techniques employed in sound feature extraction can be found in [17], of which Spectrum-Temporal Parameters (STPs) [18], Linear Prediction Coding (LPC) coefficients [19], Linear Frequency Cepstral Coefficients (LFCC) [20], Pseudo Wigner-Ville Transform (PWVT) [21], and entropy coefficients [22] are of note.

Nevertheless, the Mel-Frequency Cepstral Coefficients (MFCC) [23] are probably the most widely employed set of features in sound characterization and the majority of the sound processing applications mentioned above are based on their use. Additionally, these features have also been successfully employed in other fields, such as analysis of electrocardiogram (ECG) signals [24], gait analysis [25,26], and disturbance interpretation in power grids [27].

On the other hand, the processing and classification of anuran calls have attracted the attention of the scientific community for biological studies and as indicators of climate change. This taxonomic group is regarded as an outstanding gauge of biodiversity. Nevertheless, frog populations have suffered a significant decrease in the last years due to habitat loss, climate change and invasive species [28]. So, the continual monitoring of frog populations is becoming increasingly important to develop adequate conservation policies [29].

It should be mentioned that the system of sound production in ectotherms is strongly affected by the ambient temperature. Therefore, the temperature can significantly influence the patterns of calling songs by modifying the beginning, duration, and intensity of calling episodes and, thus, the anuran reproductive activity. The presence or absence of certain anuran calls in a certain territory, and their evolution over time, can therefore be used as an indicator of climate change.

In our previous work, several classifiers for anuran calls are proposed that use non-sequential procedures [30] or temporally-aware algorithms [31], or that consider score series [32], mainly using a set of MPEG-7 features [33]. MPEG-7 is an ISO/IEC standard developed by MPEG (Moving Picture Experts Group). In [34], the comparison of MPEG-7 and MFCC are undertaken both in terms of classification performance and computational cost. Finally, the optimal values of MFCC options for the classification of anuran calls are derived in [35].

State of the art classification of sound relies on Convolutional Neural Networks (CNN) that take input from some form of the spectrogram [36] or even the raw waveform [37]. Moreover, CNN deep learning approaches have also been used in the identification of anuran sound [38]. In spite of that, studying and optimizing the process of extracting MFCC features is of great interest at least for three reasons. First, because sound processing goes beyond the classification task, including procedures such as compression, segmentation, semantic description, sound database retrieval, etc. Secondly, because the spectrograms that feed the state-of-the-art deep CNN classifiers can be constructed using MFCC [39]. And finally due to the fact that CNN classifiers based on spectrograms or raw waveforms require intensive computing resources which makes them unsuitable for implementation in low-cost low-power-consumption distributed nodes, as is the usual case in environmental monitoring networks [35].

As presented in greater detail later, the MFCC features are a representation of the sounds in the cepstral domain. They are derived after a first integral transform (from time to frequency domain), which obtains the sound spectrum, and then a second integral transform is carried out (from frequency to cepstral domain). In this paper, we will show that, by exploiting the symmetry of the sound spectra, it is possible to obtain a more accurate representation of the anuran calls and the derived features will therefore more precisely reflect the sound.

The main contribution of the paper is to offer a better understanding of the reason (symmetry) that justify and quantify why Discrete Cosine Transform (DCT) has been extensively used to compute MFCC. In more detail, the paper will show that DCT-based sound features yielded to a significantly lower error representing spectra, which is a very convenient result for several applications such as sound compression. Additionally, through the paper it will be demonstrated that symmetry-based features (DCT) are less correlated, which is an advantage to be exploited in later classification algorithms.

## 2. Materials and Methods

### 2.1. Extracting MFCC

The process of extracting the MFCC features from the $n$ samples of a certain sound requires 7 steps in 3 different domains, which are depicted in Figure 1, and can be summarized as follows:

1.  Pre-emphasis (time domain): The sound's high frequencies are increased to compensate for the fact that the Signal-to-Noise Ratio (SNR) is usually lower at these frequencies.

2.  Framing (time domain): The *n* samples of the full-length sound segment are split into frames of short duration (*N* samples, $N \ll n$). These frames are commonly obtained using non-rectangular overlapping windows (for instance, Hamming windows [40]). The subsequent steps are executed on the *N* samples of each frame.

3.  Log-energy spectral density (spectral domain): Using the Discrete Fourier Transform (DFT) or its faster version, the Fast Fourier Transform (FFT), the *N* samples of each frame are converted into the *N* samples of an energy spectral density, which are usually represented in a log-scale.

4.  Mel bank filtering (spectral domain): The *N* samples of each frame's spectrum are grouped into *M* banks of frequencies, using *M* triangular filters centred according to the mel scale [41] and the mel Filter Bank Energy (mel-FBE) is obtained.

5.  Integral transform (cepstral domain): The *M* samples of the mel-FBE (in the spectral domain) are converted into *M* samples in the cepstral domain using an integral transform. In this article, it will be shown that the exploitation of the symmetry of the DFT integral transform obtained in step 3 yields a cepstral integral transform with a better performance.

6.  Reduction of cepstral coefficients (cepstral domain): The *M* samples of the cepstrum are reduced to *C* coefficients by discarding the least significant coefficients.

7.  Liftering (cepstral domain): The *C* coefficients of the cepstrum are finally lifted to compensate for the fact that high quefrency coefficients are usually much smaller than their low quefrency counterparts.

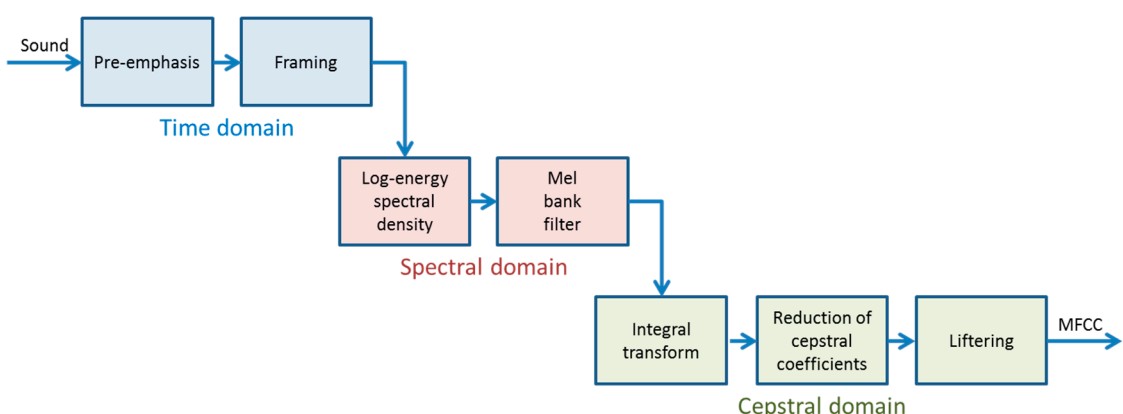

**Figure 1.** The process of extracting the Mel-Frequency Cepstral Coefficients (MFCC) features from a certain sound.

In this process, integral transforms are used twice: in step 3 to move from the time domain into the spectral domain; and in step 5 to move forward into the cepstral domain. In this paper, the symmetric properties of the DFT integral transform in step 3 will be exploited for the selection of the most appropriate integral transform required in step 5.

*2.2. Integral Transforms of Non-Symmetric Functions*

As detailed in the previous subsection, a sound spectrum is featured in order to obtain the MFCC of a sound, specifically by characterizing the logarithm of its energy spectral density. In short, this would be a particular case of the characterization of a function $f(x)$ by means of a reduced set of values where, in this case, $f(x)$ is the spectrum of a sound. To address this problem, which is none other than that of the compression of information, several techniques have been proposed, from among which the frequency representation of the function stands out. In effect, the idea underlying this type of technique is to consider the original signal, expand it in Fourier series, and then approximate the function by means of a few terms of its expansion. Thus, instead of having to supply the values of the

function corresponding to each value of $x$, only the amplitude values (and eventually also the phase) of a reduced number of harmonics are provided.

Let us consider an arbitrary example function $f(x)$, such as that shown in Figure 2, of which we know only one fragment in the interval $[x_0, x_0 + P]$ (dashed line). Now let us consider that this function is sampled, and the values only at specific points for $x = x_n$, separated at intervals $\Delta x$, are known. By denoting $N$ as the total number of points (samples) in a period, we know that $\Delta x = P/N$. The sampled function will be called $\hat{f}(x_n) = f_n$ where the hat (ˆ) above $f$ represents a sampled function.

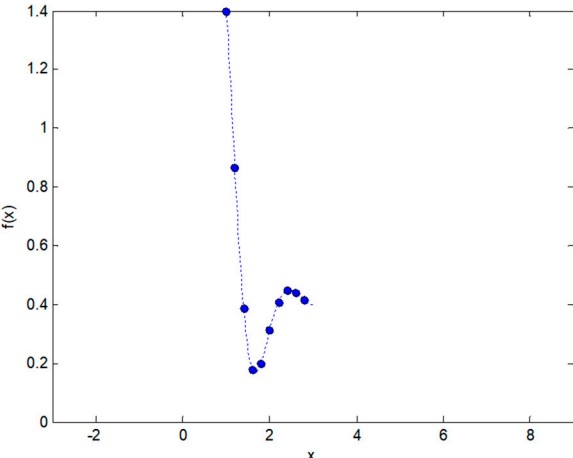

**Figure 2.** Known fragment of an example function $f(x)$ (dashed line) and its corresponding sampled function $\hat{f}(x_n)$ (dots).

The usual way to obtain the spectrum of that function is to define a periodic function $f_p(x)$ of period $P$ that coincides with the previous function in the known interval (see Figure 3), and to proceed to compute the spectrum of that new function. The spectral representation of the function $f_p(x)$ is composed of the complex coefficients of the Fourier series expansion given by [42].

$$c_k = \frac{1}{P} \int_{x_0}^{x_0+P} f(x) e^{-j\frac{2\pi k x}{P}} dx. \tag{1}$$

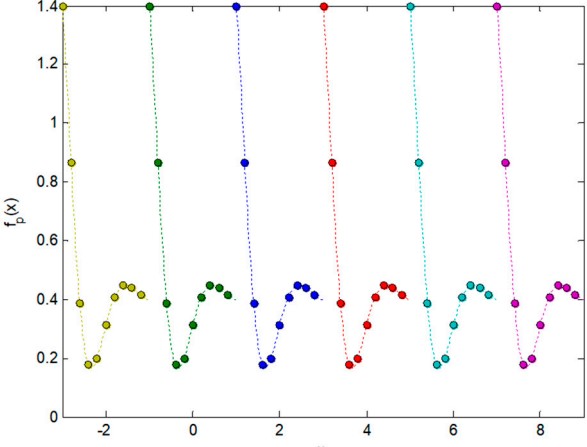

**Figure 3.** Periodic function $f_p(x)$ obtained by repetition of the known fragment of $f(x)$.

On the other hand, the sampled function, $\hat{f}(x_n) = f_n$, will have a spectral representation $\hat{c}_k$ that corresponds to $c_k$, when the sampling of the variable $x$ is taken into account. Now let us call $I(x)$ the integrand of Equation (1), i.e.,

$$I(x) = f(x)e^{-j\frac{2\pi kx}{P}},$$  (2)

and hence the spectral representation of the non-sampled function $f_p(x)$ is featured by the coefficients

$$c_k = \frac{1}{P}\int_{x_0}^{x_0+P} I(x)dx.$$  (3)

in order to obtain the values $\hat{c}_k$ that take into account the sampling of the variable $x$, the continuous calculation of the area that supposes the integral of the previous expression is substituted with the sum of the rectangles corresponding to the discrete values (sum of Riemann). In Figure 4, the calculation of the real part of $\hat{c}_1$ is depicted for the example function $f_p(x)$.

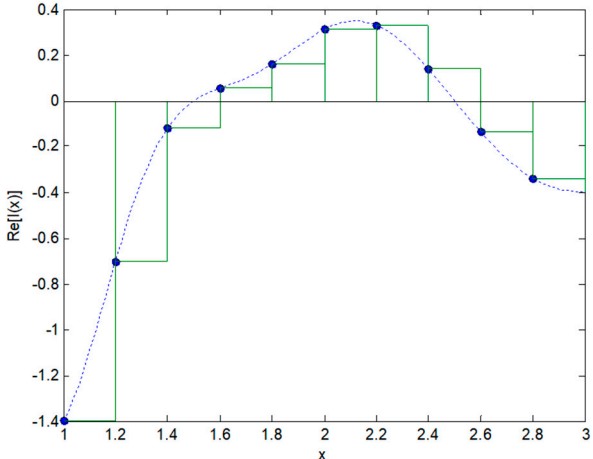

**Figure 4.** Integration of sampled functions (sum of Riemann).

Therefore,

$$\hat{c}_k \equiv [c_k]_{x=x_n} = \left[\frac{1}{P}\int_{x_0}^{x_0+P} I(x)dx\right]_{x=x_n}.$$  (4)

From this equation it can be derived (see supplementary material) that

$$\hat{c}_k = \frac{1}{N}e^{-j\frac{2\pi kx_0}{N\Delta x}}\sum_{n=0}^{N-1} f_n\, e^{-j\frac{2\pi kn}{N}}.$$  (5)

It can be observed that the spectral representation $\hat{c}_k$ depends on the point $x_0$ selected as the origin of coordinates, due to the factor $e^{-j\frac{2\pi kx_0}{N\Delta x}}$. This factor does not affect the amplitude spectrum (since its modulus is 1), but it does affect the phase spectrum corresponding to the known time-shift property of the Fourier Transform. For practical purposes, the origin of coordinates is usually considered to be the starting point of the sequence, that is, at $x_0 = 0$, and hence the spectral representation finally becomes

$$\hat{c}_k = \frac{1}{N}\sum_{n=0}^{N-1} f_n\, e^{-j\frac{2\pi kn}{N}}.$$  (6)

This expression coincides with the usual definition of the Discrete Fourier Transform (DFT) [43]. In other words: The Discrete Fourier Transform of a known fragment of a function presupposes the periodic repetition of that fragment.

### 2.3. Integral Transforms of Symmetric Functions

Let us now again consider the function $f(x)$ of which we know only sampled values of a fragment $f_n$ in the interval $[x_0, x_0 + P]$, as shown in Figure 2. An alternative way of representing its spectrum to that of periodically repeating the values $f_n$ as in Figure 3, lies in defining a sequence of values $g_n$ of length $2P$ that coincides with $f_n$ in the interval $[x_0, x_0 + P]$, which is its symmetric in the interval $[x_0 - P, x_0]$, as depicted in Figure 5.

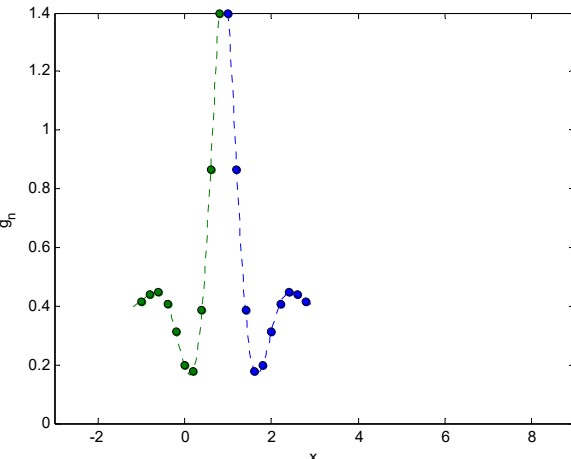

**Figure 5.** Known fragment of a symmetric example function $g(x)$ (dashed line) and its corresponding sampled function $\hat{g}(x_n)$ (dots). These functions are obtained by considering the original fragment of the example function $f(x)$ (blue) and its symmetric (green).

It can be observed that

$$\begin{aligned} g_n &= f_n \ \forall n \in [0, N-1] \\ g_n &= f_{-n-1} \ \forall n \in [-N, -1] \end{aligned}. \tag{7}$$

Subsequently, a sequence of periodic values $h_n$ of period $P' = 2P$ is defined that coincides with $g_n$ in the interval $[x_0 - P, x_0 + P]$, as shown in Figure 6.

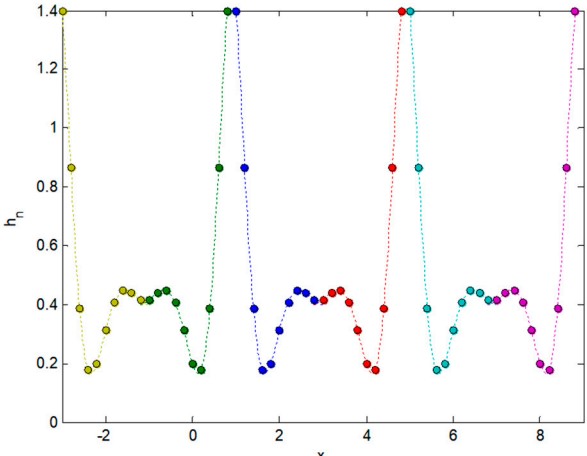

**Figure 6.** Periodic function $h_n$ obtained by repetition of the known fragment of $g_n$.

In order to obtain the spectrum of the sequence of values $h_n$ it can be written that

$$\hat{c}_k = \frac{1}{P'} \sum_{x_n = x_0 - P}^{x_n = x_0 + P - \Delta x} h_n \, e^{-j \frac{2\pi k x_n}{P'}} \Delta x. \tag{8}$$

From this equation it can be derived (see supplementary material) that

$$\hat{c}_k = \frac{1}{2N} e^{-j\frac{\pi k x_0}{N\Delta x}} \left[ e^{j\frac{\pi k}{N}} \sum_{n=0}^{N-1} f_n \, e^{j\frac{\pi kn}{N}} + \sum_{n=0}^{N-1} f_n \, e^{-j\frac{\pi kn}{N}} \right]. \tag{9}$$

As can be observed, due to the factor $e^{-j\frac{2\pi k x_0}{N\Delta x}}$, the spectral representation $\hat{c}_k$ depends on the point $x_0$ where the origin of coordinates is defined. This factor does not affect the amplitude spectrum (since its modulus is 1), but it does affect the phase spectrum, which corresponds to the known time-shifting property of the Fourier transform. For practical purposes, the origin of coordinates is usually considered to be located the midpoint of the symmetric sequence $g_n$, that is, $x_0 = \Delta x/2$, as shown in Figure 7.

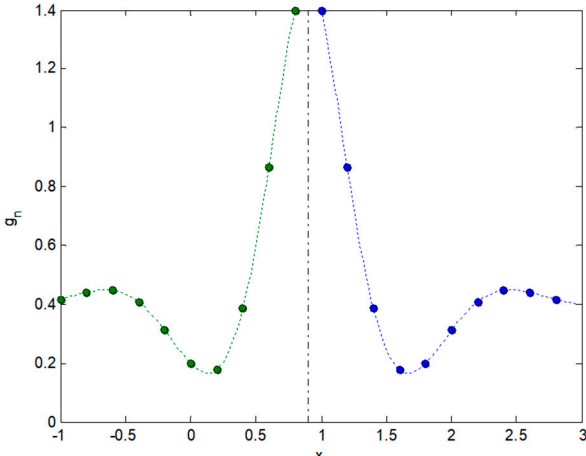

**Figure 7.** Defining the origin of coordinates.

Finally, the spectral representation becomes (see supplementary material)

$$\hat{c}_k = \frac{1}{N} \sum_{n=0}^{N-1} f_n \cos\left[ \frac{\pi k}{N}\left(n + \frac{1}{2}\right) \right]. \tag{10}$$

This expression coincides with the usual definition of the Discrete Cosine Transform (DCT) [44]. In other words, the Discrete Cosine Transform of a known fragment of a function presupposes the periodic repetition of that fragment and its symmetric.

### 2.4. Representing Anuran Call Spectra

With this digression, we can now address the question posed at the beginning of Section 2.2 concerning the best way to characterize the spectrum of a sound by using the sum of its harmonics. Note that it is necessary to compute the spectrum (step 5) of a spectrum (step 4), that is, the trans-spectrum or the cepstrum, as previously discussed. The decision regarding whether this trans-spectrum (cepstrum) should be derived using either the Fourier transform, or the cosine transform, is based on the form of the fragment $f_n$ (in this case the spectral values of the sound). That is, it should be considered whether the best approximation to the spectrum is either a periodic repetition of $f_n$ or, in contrast, a periodic repetition of $f_n$ and its symmetric.

Although this is a general question, we have addressed it in the context of a specific application by featuring anuran calls for their further classification. The dataset employed contains 1 hour and 13 minutes of sounds which have been recorded at five different locations (four in Spain, and one in Portugal) [32] and they were subsequently sampled at 44.1 kHz. The recordings include 4 types of

anuran calls and, since they have been taken in their natural habitat, are affected by highly significant surrounding environmental noise (such as that of wind, water, rain, traffic, and voices).

In this paper, the duration of the frames (step 2) was set to 10 ms, such that each frame has $N = 441$ data points and a total of $W = 434, 313$ frames are considered. The log-energy spectral density (step 3) is obtained using a standard FFT algorithm, which obtains a spectrum with $N = 441$ values. The mel-scaling (step 4) employs a set of $M = 23$ filters, and hence the mel-FBE spectrum is characterised by this number of values ($M = 23$). In step 5, two different approaches for obtaining the cepstrum are used and compared: DFT and DCT. The results are then analysed for a different number of cepstral coefficients ($1 \leq C \leq M$).

In order to carry out a more systematic study of the spectrum approximation error, let us call $E_i(n)$ the original mel-FBE spectrum of the $i$-th frame (the result of step 4), where $n$ is the filter index (equivalent to the frequency in mel scale). Let us also call $H_i(m)$ the spectrum of $E_i(n)$, that is, the cepstrum as obtained in step 5, where $m$ is the cepstral index (equivalent to the quefrency in mel scale). It can be written that $H_i(m) = \mathcal{F}[E_i(n)]$, where $\mathcal{F}$ represents either the DFT or the DCT Fourier expansions.

After reducing the number of cepstral coefficients to a value of $C \leq M$, the resulting approximate cepstrum (step 6) will be called $\widetilde{H}_i(m)$, where the tilde ($\widetilde{\ }$) above the $H$ represents an approximation. Using these $C$ values in the corresponding Fourier expansion leads to an approximation of the mel-FBE, that is, $\widetilde{E}_i(n) = \mathcal{F}^{-1}\left[\widetilde{H}_i(m)\right]$. The approximation error for the $i$-th frame is therefore $\varepsilon_i(n) = E_i(n) - \widetilde{E}_i(n)$, that is, a different error for each value of $n$, the filter index (or frequency in mel-scale). An error measure for the overall spectrum of the $i$-th frame can be obtained using the Root Mean Square Error ($RMSE_i$) defined as:

$$RMSE_i \equiv \sqrt{\frac{1}{M} \sum_{n=0}^{M-1} [\varepsilon_i(n)]^2} = \sqrt{\frac{1}{M} \sum_{n=0}^{M-1} \left[E_i(n) - \widetilde{E}_i(n)\right]^2}. \tag{11}$$

In this paper, an arbitrary selected single frame is first considered, mainly for illustration purposes. Its time-domain representation is depicted in Figure 8A while its spectrum is plotted in Figure 8B. Some other examples can be found in [32].

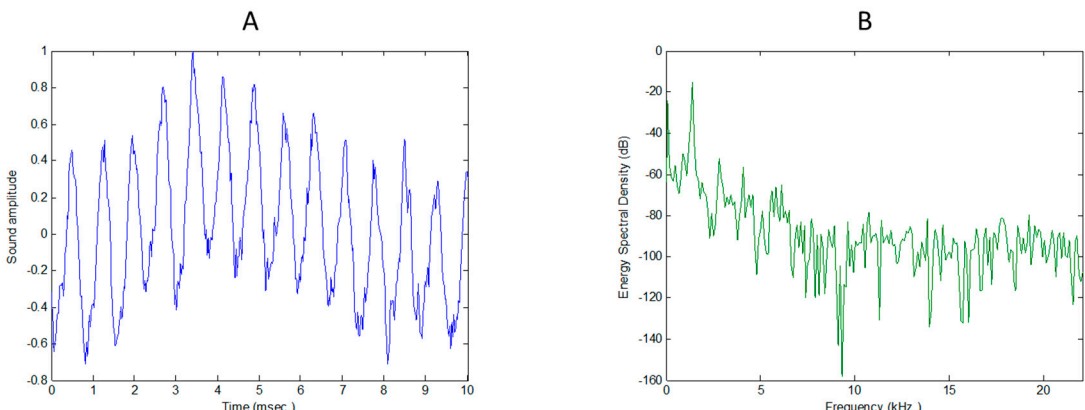

**Figure 8.** Sound amplitude for an arbitrarily selected frame of an anuran call (**A**); and its log-scale Energy Spectral Density (**B**).

Additionally, in order to compare the performance of the 2 competing algorithms obtaining the cepstrum, an overall metric for the whole dataset is considered and defined as the mean RMSE for every frame, that is,

$$RMSE \equiv \frac{1}{W} \sum_{i=1}^{W} RMSE_i = \frac{1}{W} \sum_{i=1}^{W} \sqrt{\frac{1}{M} \sum_{n=0}^{M-1} [\varepsilon_i(n)]^2}. \tag{12}$$

## 3. Results

Let us first consider a single frame, arbitrarily selected from the whole sound dataset. Although these results are limited to that specific sound frame, very similar results are obtained if a different frame is selected. Moreover, at the end of this section, the overall sound dataset is considered.

For the case of the single frame, the mel-FBE spectrum obtained in step 4 is depicted in Figure 9. This is the $f(x)$ function whose spectrum (cepstrum in this case) must be computed in step 5.

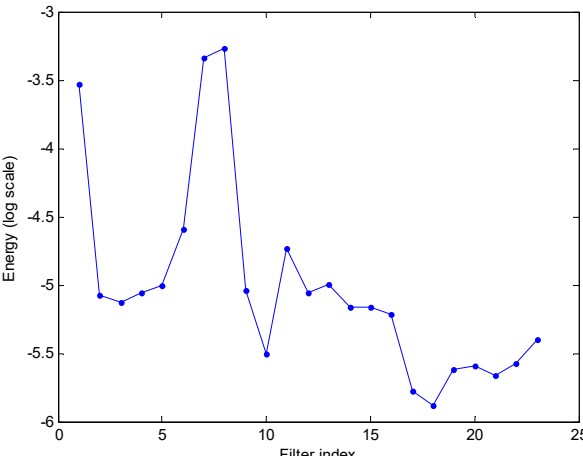

**Figure 9.** Mel Filter Bank Energy (mel-FBE) spectrum for an arbitrarily selected frame of an anuran call.

For this frame, let us consider whether it is better to use either a DFT or a DCT. The decision depends on whether the function $f(x)$ can be considered as a fragment of a periodic repetition of: (A) the fragment, as shown in Figure 10A, or (B) the function and its symmetric, as shown in Figure 10B. In the first case, the DFT should be more appropriate, while in the second case the DCT would obtain better results.

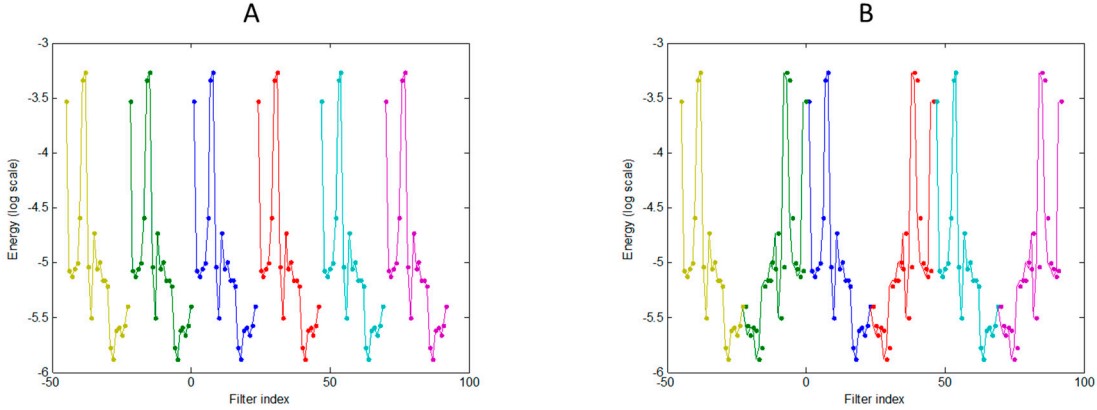

**Figure 10.** Periodic repetition of the mel-FBE spectrum (**A**); and the mel-FBE spectrum and its symmetric (**B**).

However, the mel-FBE is nothing but a rescaled and compressed way of presenting a spectrum. On the other hand, it is a well-known fact that the spectrum of a real signal is symmetric with respect to the vertical axis [43]. And finally, it is also known that the spectrum of a sampled signal is periodic [45]. For this reason, the repetition of the fragment of Figure 9 corresponds to Figure 10B and, therefore, using the DCT to compute its trans-spectrum (or cepstrum) should obtain better results. This hypothesis is verified in the following paragraphs for the selected frame, and, later in this section, it is verified for the whole dataset.

The number of coefficients obtained by applying either DCT or DFT is $M = 23$, that is, they have the same number of values that define the mel-FBE. The resulting cepstrum for the selected frame is shown in Figure 11.

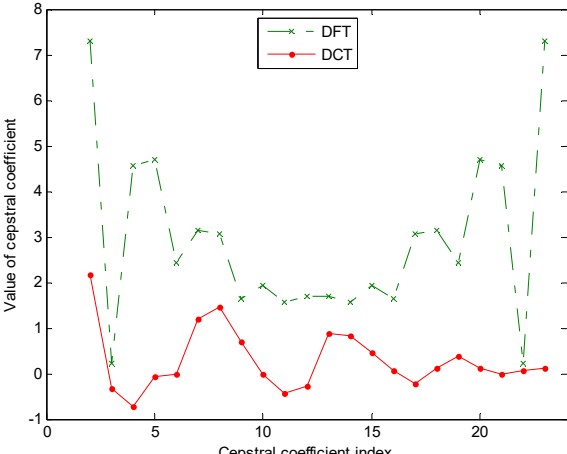

**Figure 11.** Cepstral representation of the mel-FBE spectrum (cepstrum).

The ability to compress information of the Fourier transforms (either in the DFT or DCT version) lies in the fact that it is not necessary to consider the full set of the $M$ coefficients of the Fourier expansion to obtain a good approximation of the original function. In Figure 12, the original mel-FBE spectrum is depicted for the example frame, and those spectra recovered using $C \leq M$ cepstral coefficients obtained using DCT.

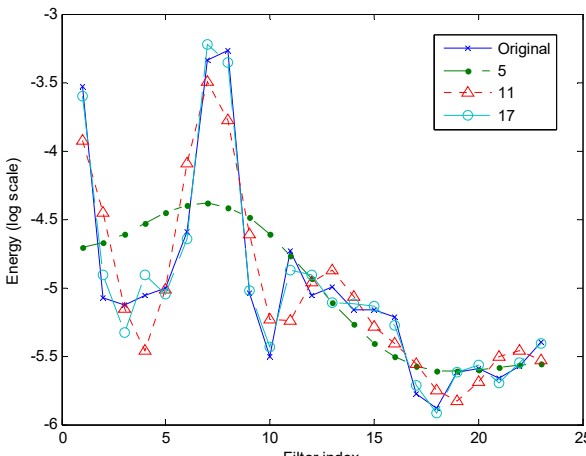

**Figure 12.** Mel-FBE spectrum for an arbitrarily selected frame of an anuran call. Original spectrum and recovered spectra using a different number of Discrete Cosine Transform (DCT) cepstral coefficients.

Additionally, as expected, the DCT achieves approximations to the original spectrum that are, in general, significantly better than those obtained for the DFT with the same number of coefficients. In Figure 13, the original mel-FBE spectrum is depicted for the example frame, and those spectra recovered using $C = 11$ cepstral coefficients obtained using DFT and DCT.

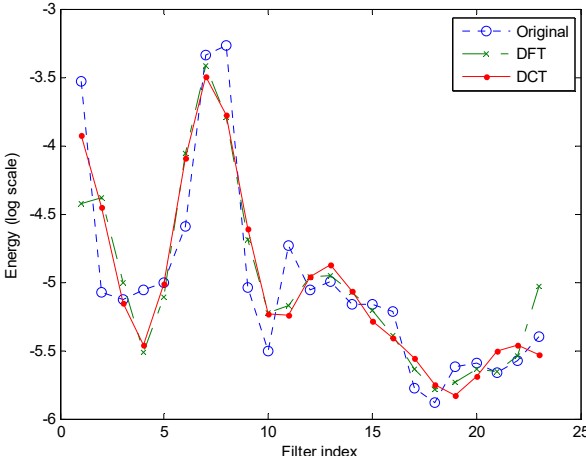

**Figure 13.** Mel-FBE spectrum for an arbitrarily selected frame of an anuran call. Original spectrum and recovered spectrum using $C = 11$ coefficients obtained using Discrete Fourier Transform (DFT) and DCT.

In order to quantify the error of recovering the selected mel-FBE spectrum using $C \leq M$ cepstral coefficients, the Root Mean Square Error (RMSE) is computed in accordance with Equation (11). The value of RMSE as a function of the number $C$ of cepstral coefficients used for the recovery of the spectrum is depicted in Figure 14, both for DFT and DCT.

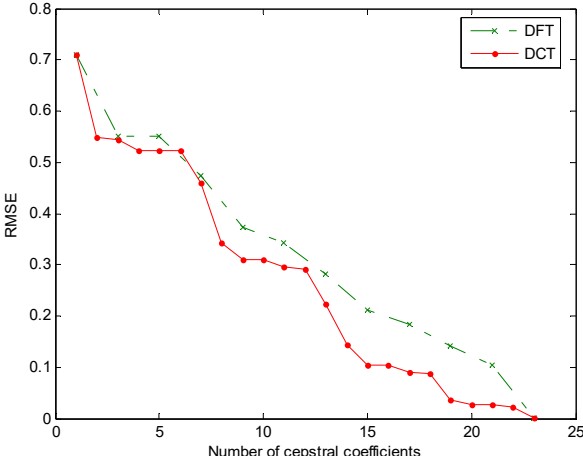

**Figure 14.** Root Mean Square Error recovering the original mel-FBE spectrum when a different number of $C$ cepstral coefficients are used. The cepstral coefficients are obtained applying either DFT or DCT.

This analysis can be extended to include the computation of the RMSE for the whole dataset in accordance with Equation (12). The value of RMSE as a function of the number $C$ of cepstral coefficients used for the recovery of the spectrum is depicted in Figure 15 for DFT and DCT separately.

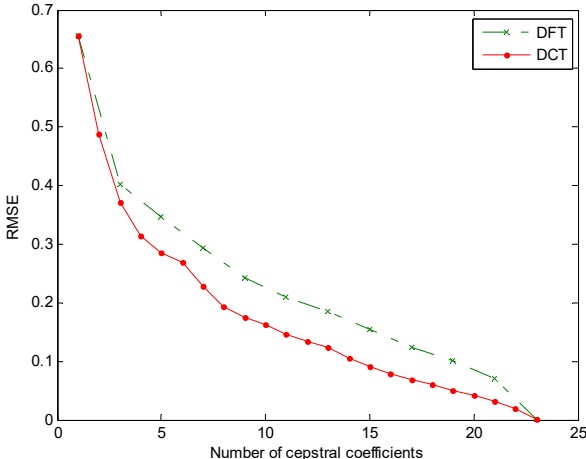

**Figure 15.** Root Mean Square Error for the whole dataset when either DFT or DCT is employed.

## 4. Discussion

Let us first consider the $RMSE_i$ for a single frame as depicted in Figure 14. Let us now regard the case where, for instance, the number of values required to describe the mel-FBE spectrum ($M = 23$) is halved, and hence the number of cepstral coefficients used for the recovering an approximation of the spectrum is $C = 11$ (in accordance with Equations (6) and (10)).

In this case, it can be observed that $RMSE_i$ is 0.34 for DFT, and 0.30 for DCT. On the other hand, as depicted in Figure 9, the values of the mel-FBE spectrum lie within the range $[-6, -3]$, with a mean value of $-5.02$. This means that the relative error of the spectrum representation is only 6.84% for DFT (5.36% for DCT) when the number of values employed for that representation are halved.

Let us now focus on the RMSE when the DFT is used (green line), either for a single frame (Figure 14) or for the whole dataset (Figure 15). In both cases, it can be observed that RMSE has values only for an odd number of cepstral coefficients. This fact can be explained by recalling that, according to Equation (6), every DFT cepstral coefficient $\hat{c}_k$ is a complex number for $1 \leq k \leq M - 1$ and a real number for $k = 0$. On the other hand, according to Equation (10), the DCT cepstral coefficients $\hat{c}_k$ are real numbers for every value of $k$. Additionally, it has to be considered that DFT cepstrum is symmetric (green line in Figure 11). Therefore, for $k > 0$, it can be written that $\hat{c}_k = \hat{c}_{M-k+1}$ and, therefore, only one of these 2 terms have to be kept for recovery purposes. These circumstances jointly explain the odd number of DFT cepstral coefficients.

To clarify this idea, let us consider an example where $M = 23$ and $C = 5$. The DCT cepstrum is then described using $\hat{c}_0$, $\hat{c}_1$, $\hat{c}_2$, $\hat{c}_3$ and $\hat{c}_4$, that is, 5 real numbers which can be employed to approximately recover the mel-FBE spectrum. On the other hand, the DFT cepstrum is described using $\hat{c}_0$, which is a real number, and $\hat{c}_1$ and $\hat{c}_2$, which are complex numbers, that is, although 3 terms are used, a total of 5 values (coefficients) are required. However, to approximately recover the mel-FBE spectrum, the terms $\hat{c}_0$, $\hat{c}_1$, $\hat{c}_2$, $\hat{c}_{23}$ and $\hat{c}_{22}$ can be used since $\hat{c}_1 = \hat{c}_{23}$ and $\hat{c}_2 = \hat{c}_{22}$.

As regards the results obtained for the whole dataset (Figure 15), it can be seen that DCT is better at describing the mel-FBE spectra than is its DFT counterpart. This improvement (decrease of the RMSE), can be measured by defining $\Delta RMSE \equiv RMSE_{DFT} - RMSE_{DCT}$ (Figure 16A) or its relative value $\Delta RMSE(\%) \equiv 100 \cdot \Delta RMSE / RMSE_{DFT}$ (Figure 16B). For example, for $C = 11$, the RMSE is reduced from 0.209 (DFT) to 0.146, which involves an improvement of approximately 30%. For the degenerated cases where $C = 1$ and $C = M$, there is no improvement. In the first case, only $\hat{c}_0$ is used which, according to Equations (6) and (10), is the mean value of the mel-FBE spectrum, that is, the DFT and DCT recovering methods have the same error. On the other hand, if $C = M$ then no reduction on the number of coefficients is achieved, and both equations exactly recover the original spectrum (no error).

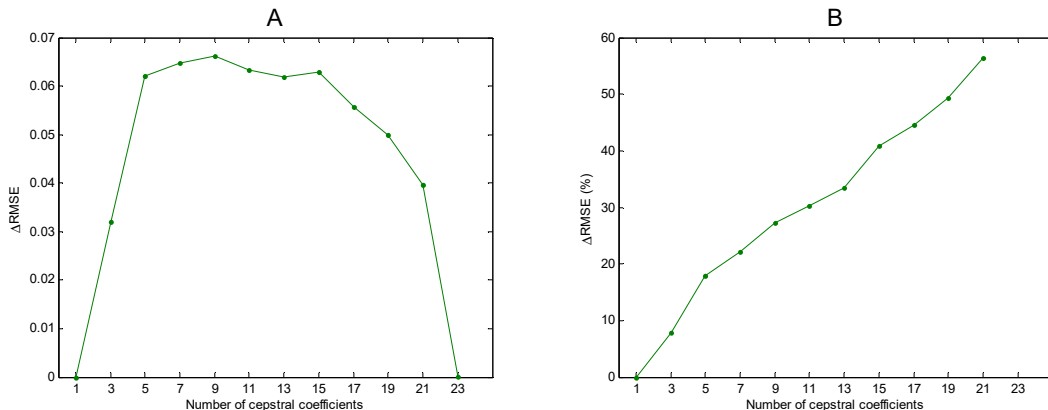

**Figure 16.** Improvement of DCT over DFT describing mel-FBE spectra. (**A**): $\Delta RMSE$. (**B**): $\Delta RMSE(\%)$.

The above results concern the mean improvement of DCT over DFT for every frame in the dataset. In a more in-depth analysis, let us also compute its probability density function (pdf). The results are depicted in Figure 17. In panel A, the pdf is shown for several values of the number of cepstral coefficients (*C*). In panel B, the value of the pdf is colour-coded as a function of the improvement ($\Delta Error$) and of the number of cepstral coefficients (*C*). It can be observed that only a negligible number of the frames present a significant negative improvement, thereby demonstrating that DCT is superior to DFT.

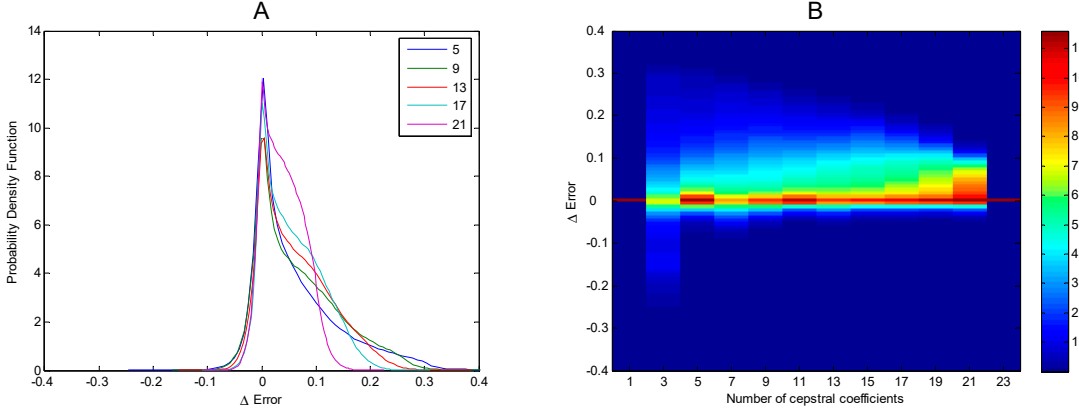

**Figure 17.** Improvement of DCT over DFT in describing mel-FBE spectra. (**A**): Probability density function for several values of the number of cepstral coefficients. (**B**): Probability density function for each value of the number of cepstral coefficients.

The higher performance of DCT over DFT is due to the fact that the mel-FBE spectra are a special type of function derived from symmetric sound spectra. Consequently, if DCT and DFT were compared in the task of recovering arbitrary functions, they would each present equal performance. To demonstrate this claim, one million *M*-value arbitrary functions are randomly generated ($M = 23$), and DFT and DCT are then employed to recover the original function with a reduced set of *C* coefficients to measure the errors of that recovery. Finally, the improvement of DCT over DFT is computed. The results are depicted in Figure 18 where it can be observed that positive and negative improvements are symmetrically distributed around a zero-mean improvement. Therefore, it can be concluded that DCT and DFT have similar performance in describing arbitrary functions.

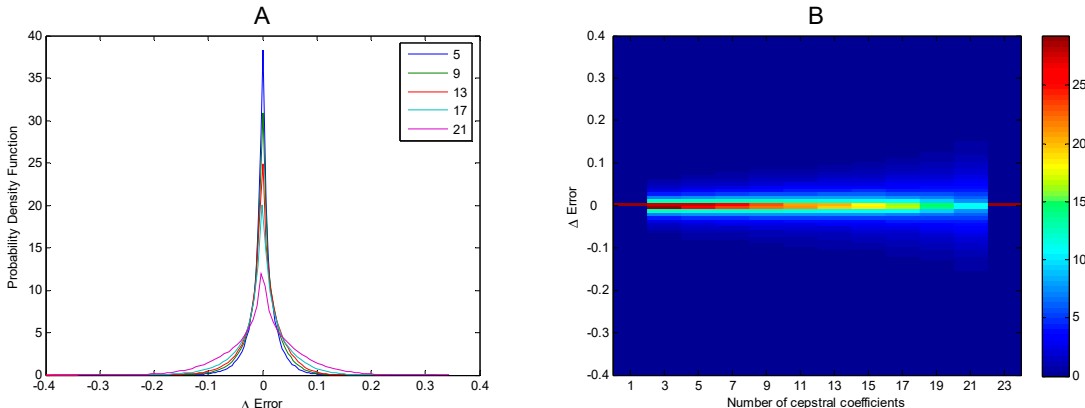

**Figure 18.** Improvement of DCT over DFT in describing arbitrary function. (**A**): Probability density function for several values of the number of cepstral coefficients. (**B**): Probability density function for each value of the number of cepstral coefficients.

From the above results, it is clear that DCT offers superior performance featuring mel-FBE spectra and, therefore offers superior performance featuring sounds. When the purpose of these features is to be used as input to some kind of classifier, then DCT offers an additional advantage. It is a well-established result that classifiers obtain better results if their input features are low-correlated. The reason is clear: a classification algorithm that includes a new feature that is highly correlated with previous features adds almost no new information and, therefore, almost no classification improvement should be expected. Let us therefore examine the correlation between coefficients obtained by DFT and those by DCT.

Let us call $\mu_u$ the mean value of the $u$-th coefficient $\hat{c}_{ui}$ describing the $i$-th frame, obtained by

$$\mu_u = \frac{1}{W} \sum_{i=1}^{W} \hat{c}_{ui}, \tag{13}$$

where $W$ is the total number of frames in the dataset. The variance $\sigma_u^2$ of the $u$-th coefficient can be obtained by

$$\sigma_u^2 = \frac{1}{W-1} \sum_{i=1}^{W} (\hat{c}_{ui} - \mu_u)^2. \tag{14}$$

The correlation $\rho_{uv}$ between the $u$-th and the $v$-th coefficient for the whole dataset is therefore given by

$$\rho_{uv} = \frac{1}{W-1} \sum_{i=1}^{W} \frac{\hat{c}_{ui} - \mu_u}{\sigma_u} \cdot \frac{\hat{c}_{vi} - \mu_v}{\sigma_v}. \tag{15}$$

In Figure 19, the absolute values of the correlation are shown, whereby the values for the case $M = 23$ are colour-coded. The correlations corresponding to the DFT are shown in panel A and those corresponding to DCT in panel B. In the DFT case, each $\hat{c}_{ui}$ factor is a complex number, and hence the total number of values is 46, whereby the first 23 coefficients represent the real parts and the last 23 the imaginary parts. By simply considering the colours in that figure, it is clear that DCT coefficients are less correlated.

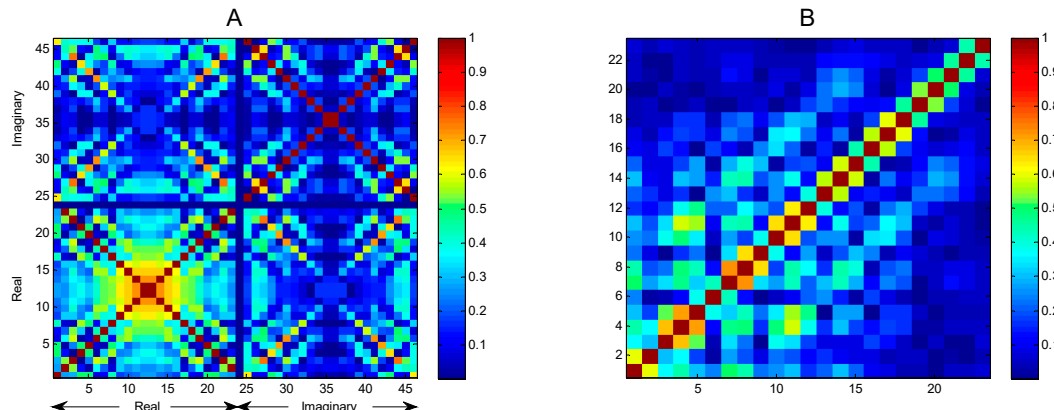

**Figure 19.** Correlation between cepstral coefficients describing mel-FBE spectra for DFT (panel **A**) and DCT (panel **B**).

An alternative way to present this result is by using a histogram of the values of the correlation coefficients, as depicted in Figure 20. Those corresponding to DCT are more frequent for the low values of correlation, that is, DCT-obtained features are less correlated than those obtained using DFT. Hence, classifiers of a more efficient nature should be expected from using DCT.

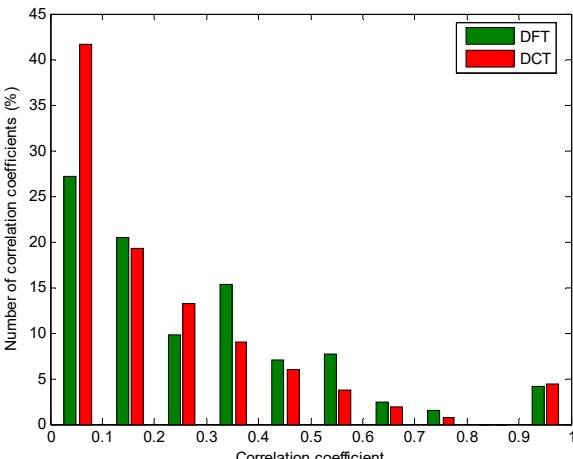

**Figure 20.** Histogram of the correlation among cepstral coefficients describing mel-FBE spectra for DFT and DCT.

When the MFCC features are used as input of a later classification algorithm, the lower correlation of DCT-obtained features should yield to a better classification performance. The results obtained classifying anuran calls [35] do confirm a slight advantage for the DCT as it is reflected in Table 1. This table has been produced taking the best result (geometric mean of sensitivity and specificity) obtained through a set of ten classification procedures: minimum distance, maximum likelihood, decision trees, k-nearest neighbors, support vector machine, logistic regression, neural networks, discriminant function, Bayesian classifiers and hidden Markov models.

**Table 1.** Classification performance metrics for DCT and DFT.

| Cepstral Transform | ACC | PRC | $F_1$ |
|---|---|---|---|
| DFT | 94.27% | 74.46% | 77.67% |
| DCT | 94.85% | 76.76% | 78.93% |

Let us finally consider the computing efforts required for these two algorithms which mainly depend on the number of samples defining the mel-FBE spectra. Fast versions of DFT and DCT

algorithms have been tested on a conventional desktop personal computer. The results are depicted in Figure 21. It can be seen that DCT is about one order of magnitude slower than DFT. Although this fact is certainly a drawback of DCT it has a limited impact on conventional MFCC extraction process because the number of values describing the mel-FBE spectra is usually very low (about 20). Additional studies on processing times for anuran sounds classification can be found in [34].

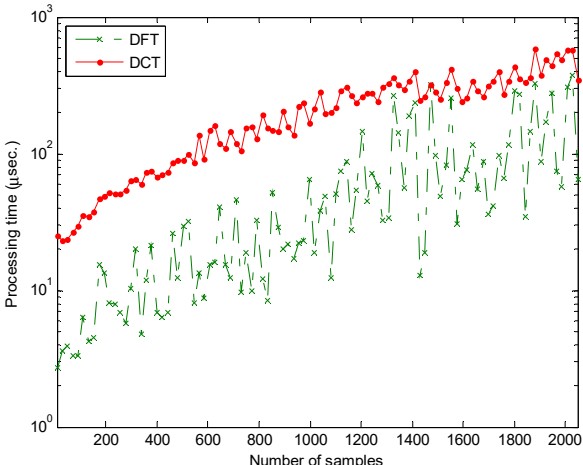

**Figure 21.** Processing time required to compute the DFT and DCT vs. the number of samples describing mel-FBE spectra.

## 5. Conclusions

In this article, it has been shown that DCT outperforms DFT in the task of representing sound spectra. It has also been shown that this improvement is due to the symmetry of the spectrum and not to any intrinsic advantage of DCT.

In representing the mel-FBE spectra required to obtain the MFCC features of anuran calls, DCT errors are approximately 30% lower than DFT errors. This type of spectra is therefore much better represented using DCT.

Additionally, it has been shown than MFCC features obtained using DCT are remarkably less correlated than those obtained using DFT. This result will make DCT-based MFCC features more powerful in later classification algorithms.

Although only one specific dataset has been analysed herein, the advantage of DCT can easily be extrapolated to include any sound since this advantage is based on the symmetry of the spectrum of the sound

**Supplementary Materials:** The following are available online at http://www.mdpi.com/2073-8994/11/3/405/s1, supplementary material: Derivation of integral transforms expressions.

**Author Contributions:** Conceptualization, A.L.; investigation, A.L., J.G.-B., A.C. and J.B.; writing—original draft, A.L., J.G.-B., A.C. and J.B.

**Funding:** This research received no external funding.

**Acknowledgments:** The authors would like to thank Rafael Ignacio Marquez Martinez de Orense (Museo Nacional de Ciencias Naturales) and Juan Francisco Beltrán Gala (Faculty of Biology, University of Seville) for their collaboration and support.

**Conflicts of Interest:** The authors declare there to be no conflict of interest.

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
