# Peer review of "Exploiting the Symmetry of Integral Transforms for Featuring Anuran Calls"

_symmetry, doi:10.3390/sym11030405_

Reviewer 1 Report

I thank the Authors for addressing my comments.

In my opinion, the scope and motivations of the manuscript are now clear and the experimental analysis is more solid. The paper reads better without the math.

I have only a minor comment. Even if there is a reference to [35], please provide some info about the classifier used to produce table 1.

Author Response

Please find the reply in the file enclosed

Reviewer 2 Report

Authors addressed all my required modifications

Author Response

(The authors gave the same response as above.)

Reviewer 3 Report

The introduction describes the problem and provides a complete state of the art regarding the extraction of cepstral coefficients of the anuran calls.

The methodology is clearly and precisely reported in section 2, providing correct links to the supplementary material to better explain the most technical steps. 

Also, the section about the experimentation and the discussion are very well written regarding the form, and accurate, regarding the content.

Also, the results obtained are remarkable.

Minors: 

eq 7 : [-1, -N] should be [-N, -1]

Author Response

Please find the reply in the file enclosed

This manuscript is a resubmission of an earlier submission. The following is a list of the peer review reports and author responses from that submission.

Round  1

Reviewer 1 Report

The paper compared the performance of using DFT and DCT techniques  in obtaining cepstral coefficients. The performance comparisons are done in the context of processing anuran calls. The obtained result is that DCT clearly outperforms DFT. 

The paper makes a good explanation of the topic and comparison. My main concern is that the paper does not propose something new. In calculation of MFCC coefficients already the DCT transform is used. The main conclusion of the paper is that the technique that is already used in calculation of MFCC coeff is better than something we don't use. The application to Anuran calls is not something that makes the application novel as well. Only contribution I see from the paper is that it makes a better understanding of why we are already using DCT coefficients in MFCC. I am not thinking that a better understanding of a topic is enough for the journal publication. 

I think the paper is well written. The language and tests are understandable. 

Reviewer 2 Report

Authors perform a methodological comparison between Discrete Fourier Transform (DFT) and Discrete Cosine Transform (DCT), both theoretical and using data sets, for the task of obtaining the cepstral coefficients for anuran calls. Authors use some well-known properties about the spectrum of real- and discrete-time signals, as its symmetry and periodicity. Before publication can be granted, authors are asked to provide some modifications and clarifications.

General comments:

-          Authors should elaborate more about the significance of anuran calls, that would help the non-specialized reader to appreciate the relevance of the proposed methods.

 -          Authors are requested to summarize the main contributions of their work at the end of introduction section.

 -          Could authors include a comparison regarding the computations task between both transforms?

 -          In page 6, line 141, should it say Discrete Fourier Transform (DFT)?

 -          In page 10, equation (38), the equal sign (=) is missing in between squared roots.

 -           In results section, could authors provide at least one anuran call signal and its corresponding frequency spectrum in Hz? That would be helpful for non-specialized readers and would help to follow the provided results.

 -          In many figures with multiple curves, e.g. Figs. 10-14, the individual curves are not distinguished when printed in grey scale. Authors are requested to used different line widths, line styles, markers, and color that help differentiate the individual curves.

Reviewer 3 Report

I do not understand what is the scope of this paper. DCT has been used for almost 50 years to compute MFCC features. 

The mathematical derivation is sections 2.2 to 2.4 is useless. This is not a tutorial on digital signal processing for master students. I think that we can reasonably expect that a reader knows the very basics of DSP.

I can accept that, given the peculiarity of the audio signals at hand (anuran calls), one may wonder whether traditional MFCC are suitable or not and would try to use DFT instead of DCT (personally I would try other things...). But, if it is the case, the experiments should involve the final classifier (as mentioned in the end). The fact that the compression introduces less errors, does not mean that the features are actually more discriminative. In addition, this analysis is a bit outdated as, today, almost all audio classification tasks are addressed using neural networks fed with logpower-spectrum, mel filter banks or even the raw waveforms.